# Hydrogel-Forming Microneedles with Applications in Oral Diseases Management

**DOI:** 10.3390/ma16134805

**Published:** 2023-07-03

**Authors:** Yuqing Li, Duohang Bi, Zhekai Hu, Yanqi Yang, Yijing Liu, Wai Keung Leung

**Affiliations:** 1Periodontology and Implant Dentistry, Faculty of Dentistry, The University of Hong Kong, Hong Kong, China; liyuqing@connect.hku.hk; 2Hubei Key Laboratory of Bioinorganic Chemistry and Materia Medica, Hubei Engineering Research Center for Biomaterials and Medical Protective Materials, School of Chemistry and Chemical Engineering, Huazhong University of Science and Technology, Wuhan 430074, China; dhbi@hust.edu.cn (D.B.); yijingliu@hust.edu.cn (Y.L.); 3Division of Paediatric Dentistry and Orthodontics, Faculty of Dentistry, The University of Hong Kong, Hong Kong, China; huzhekai@connect.hku.hk (Z.H.); yangyanq@hku.hk (Y.Y.)

**Keywords:** drug administration routes, drug delivery systems, drug implants, hydrogels

## Abstract

Controlled drug delivery in the oral cavity poses challenges such as bacterial contamination, saliva dilution, and inactivation by salivary enzymes upon ingestion. Microneedles offer a location-specific, minimally invasive, and retentive approach. Hydrogel-forming microneedles (HFMs) have emerged for dental diagnostics and therapeutics. HFMs penetrate the stratum corneum, undergo swelling upon contact, secure attachment, and enable sustained transdermal or transmucosal drug delivery. Commonly employed polymers such as polyvinyl alcohol (PVA) and polyvinyl pyrrolidone are crosslinked with tartaric acid or its derivatives while incorporating therapeutic agents. Microneedle patches provide suture-free and painless drug delivery to keratinized or non-keratinized mucosa, facilitating site-specific treatment and patient compliance. This review comprehensively discusses HFMs’ applications in dentistry such as local anesthesia, oral ulcer management, periodontal treatment, etc., encompassing animal experiments, clinical trials, and their fundamental impact and limitations, for example, restricted drug carrying capacity and, until now, a low number of dental clinical trial reports. The review explores the advantages and future perspectives of HFMs for oral drug delivery.

## 1. Introduction

“Microneedles” refer to tiny, needle-like structures that are typically less than 1 mm long and used for transdermal delivery of drugs or other therapeutic agents [1]. By penetrating the outer layer of the skin, therapeutic agent(s) can be delivered efficiently and effectively to the tissues beneath and/or to the systemic circulation in a controlled fashion, which is predetermined by the microneedle (MN) patch design, designated anatomical location of application, and material properties of all agents involved.

MNs can be shaped in a variety of ways: array formation or individual needles, while the needles can be formed by a variety of natural/synthetic materials: glass, hydrogels, metals, polymers, silicone, and zeolite. These devices are widespread across various medical fields, especially for applications where minimization of discomfort or pain is desirable, including dermatology, diabetes management, and vaccination [2,3]. In recent years, significant progress has been made in developing microneedles as a promising approach for transdermal drug delivery, such as solid MN, coated MN, hollow MN, and hydrogel MN (Figure 1).

Hydrogel-forming microneedles (HFMs) exhibit sufficient mechanical strength to penetrate the skin on the application while readily absorbing tissue fluid after implantation into the epithelium [4]. This results in the swelling of the polymer and subsequently forming continuous, unimpeded hydrogel microchannels, which open diffusion portals for drug delivery, thereby allowing the release of the therapeutic agent and the anticipated therapeutic actions in the body [5].

Concerning base/structural ingredients, HFMs could be classified as natural [6] or synthetic hydrogel MNs [7] with structural components from natural (e.g., hyaluronic acid (HA), gelatin, etc.) or synthetic polymers (e.g., polyvinyl alcohol, sodium polyacrylate, methacrylic acid polymers, etc.), respectively. Certain HFMs such as HA hydrogels and Gelatin methacryloyl (GelMA) hydrogels have received more attention due to their decent biocompatibility, degradability, and non-toxic nature, making HFMs one of the most commonly used/advocated local drug delivery protocols in biomedical applications.

Hydrogel MNs are prepared by inducing the polymerization of agents such as acrylamide, N-isopropylacrylamide, or methacrylated HA in the hydrogel, followed by injecting the fluid mixture into an MN mold, left to cure, then de-mold followed by preservation through drying [1]. Gelation of hydrogels can be achieved by a variety of mechanisms [8], including physical cross-linking of polymer chains, electrostatic interactions, and covalent chemical cross-linking, targeting the formation of a hydrophilic polymer network that (i) can absorb water and swell, (ii) has a porous structure that encapsulates the active agent such as the drug to be delivered, (iii) presents in a solid state and therefore minimizes drug loss during transport, and (iv) delivers the drug to a predetermined anatomical location or particular tissue type so as to enhance the therapeutic efficacy of the agent involved [9]. Hydrogel polymerization is often carefully physically and/or chemically modulated for specific controlled drug release protocols in designated therapeutic applications [10].

Compared with traditional drug delivery routes (e.g., oral administration, intra-muscular/-venous injections), transdermal delivery, which mainly involves placing an MN patch on the skin, puncturing the skin stratum corneum, and delivering the drug to the bloodstream through the skin tissue, is a delivery protocol of great interest because of the advantages of self-administration, on-demand drug delivery, good patient compliance, and avoidance of first-pass effect in the liver and/or drug degradation in the gastrointestinal tract [11]. HFM has a wide range of applications, from medicine to cosmetics to biotechnology. HFM has been utilized in the medical field to deliver transdermal drugs, administer vaccinations [3], monitor glucose levels [12], and treat wounds [13]. Microencapsulated cell delivery using HFM [14] has gained increasing interest in recent years. Through this innovative approach, therapeutic cells are encapsulated within hydrogel matrices and delivered through the network of microneedles. Regenerative medicine, tissue engineering, and cell-based therapies are now possible.

Similarly, the oral mucosal drug delivery system is a new type of drug delivery scheme developed in recent years, specifically targeting diseases and conditions in the mouth, which avoids gastrointestinal enzyme metabolism, acid degradation, and hepatic pre-systemic metabolism. It thus improves the bioavailability of drugs for local or systemic therapy [15]. In dentistry, the application of hydrogel-forming microneedles is gaining significant attention. Researchers and clinicians have been actively exploring the potential of these microneedles to improve oral health care, prevention, and treatment. Laboratory-based and clinical studies have been conducted to investigate using hydrogel-forming microneedles for oral disease prevention, diagnosis, and treatment. Most research in this area has been shown in the medical field, while the potential of HFM applications in dentistry remains to be explored. This paper will provide a brief overview of HFM materials and fabrication methods, medical diagnosis and disease management, the oral delivery approach, the current applications, limitations, and potential future uses in dentistry.

## 2. Materials Used for Hydrogel-Forming Microneedles: Fabrication and Characteristics

Since MNs act by penetrating the protective human skin/mucosal layer, the material must be biocompatible, i.e., not immunogenic nor foreign body reaction prone. In most cases, hydrogels are biocompatible and can be applied without causing any harm/discomfort to the host. Many of the polymers used for HFMs were previously researched and extensively trialed for use in medicine—particularly with the essential properties of biocompatibility and biodegradability (Table 1).

### 2.1. Natural Polymer Fabricated Hydrogel-Forming Microneedles

Biocompatible materials are used to fabricate MNs made of natural polymers. This type of drug delivery system uses MNs that form hydrogels. In carbohydrate-based MNs, natural polysaccharides such as chitosan, cellulose, and starch are used to create the hydrogel matrix. Protein-based MNs natural polymers could be composed of gelatin or silk fibroin as the matrix-forming component. These types of MNs are biocompatible, biodegradable, and non-toxic. Biodegradability, biocompatibility, and swelling capacity are among the properties of these MNs. In the field of drug delivery, wound healing, and transdermal vaccination, natural polymer-fabricated hydrogel-forming MNs have shown promising results.

Natural and synthetic polymer hydrogels differ in their characteristics and applications. Natural polymer hydrogels, derived from biogenic sources such as proteins or polysaccharides, exhibit inherent biocompatibility and biodegradability. These hydrogels typically possess a high water content and can faithfully replicate the extracellular matrix, providing an appropriate microenvironment for cellular growth and tissue regeneration [16]. Synthetic polymer hydrogels offer adjustable and stable matrices for drug delivery, tissue engineering, and biomedical applications. However, synthetic polymer hydrogels may necessitate additional modifications to enhance biocompatibility [17]. In summary, while natural polymer hydrogels excel in biocompatibility and mimicking the native tissue environment, synthetic polymer hydrogels provide multifunctionality and precise control over material properties, making them widely applicable in biomaterials [18]. The choice between natural and synthetic polymer hydrogels depends on the specific requirements of the desired application, balancing factors such as biocompatibility, mechanical performance, and control over functionality.

#### 2.1.1. Carbohydrate-Based Microneedles

Carbohydrates are among the oldest MN base materials [19]. Carbohydrates provide a range of functional groups that allow tunable properties and designated performance to be designed and developed. In living beings, polysaccharides support structural components, energy storage, lubrication, and inter-cellular signal transmission. The use of natural polysaccharides for pharmaceutical applications has become commonplace due to recent discoveries regarding the novel role that biopolymers play in medicine and pharmacy. One crucial aspect of polysaccharides is their biocompatibility, making them suitable for various medical applications without causing harmful effects on the human body.

In addition to biocompatibility, specific polysaccharides have been found to possess bioactive properties, such as the ability to inhibit cancer cells [20]. For example, polysaccharide-based pectin inhibits cancer cells by inducing apoptosis [21], as do extracts from *Grateloupia longifolia*, *Gracilaria lemaneiformis*, and other plants [22]. Therefore, natural polysaccharides with bioactive properties are critical biomaterials for healthcare management.

##### Chitosan or Chitosan Derivatives

The alkaline glucosamine and *N*-acetyl glucosamine copolymer chitosan (CS) is derived from chitin, which originates from lower life forms such as certain algae, fungi, invertebrates, insects, or crustaceans. This material’s superior biocompatibility, biodegradability, and antimicrobial properties make it an essential component of food and biomedical engineering [23]. CS has also been widely used in medicine, cosmetics, and water treatment applications due to its enhanced water solubility and antibacterial activity, which are attributed to the superior interaction between carboxymethyl groups and water and its inherent biocompatibility.

Chitosan hydrogel MNs have also been explored for transmucosal vaccine delivery, e.g., via the mouth or nose portal [24]. Bobbala and Hook delivered an antigen in chitosan hydrogel MNs to the oral mucosa in a rat model and elicited a robust immune response, suggesting the feasibility of HFM oral vaccine delivery [25].

Concerning oral/dental applications, the use of dental pulp stem cell-derived exosomes (DPSC-Exos) was explored [26], targeting the treatment of periodontitis, which is an oral disease condition characterized by a high proportion of proinflammatory macrophages as a result of immune responses to periodontopathogenic bacteria. DPSC-Exos was capable of suppressing periodontal inflammation and modulating immune response due to miR-1246 present in DPSC-Exos, a therapeutic potential in treating periodontitis based on a Gene Ontology term enrichment analysis. The research team demonstrated that incorporating DPSC-Exos into chitosan hydrogel (DPSC-Exos/CS) accelerates healing in mice periodontitis models, which changed the proinflammatory macrophage phenotypes to anti-inflammatory. Further research is on the way for the therapeutic agent carried by chitosan hydrogel [26].

##### Hyaluronic Acid

Hyaluronic acid (HA) is a glycosaminoglycan found as a natural constituent in epithelia, dermis, and connective tissues, including dental pulp and periodontal connective tissue. The skin contains approximately 50% of the total body HA. The human body degrades about one-third of its total HA daily and replaces the same amount with newly synthesized HA. During physiological conditions, HA is converted to its more hydrophilic sodium salt. It can hold 1000 times its own water weight because HA contains numerous hydroxyl groups. HA water retention ability enables many homeostatic functions as well as support and maintenance of diverse physiological processes in the human body. HA MNs are highly biocompatible, resistant to deformation, and are a common ingredient in skin care products [27].

Moreover, HA MNs are strong enough to penetrate the skin, readily dissolve, and release drug/active ingredients within a predetermined short period. Furthermore, no requirement of heating or organic solvents during HA MN fabrication enhances the preservation and stability of heat- and/or chemical-liable agents/drugs, such as insulin. Gill and Prausnitz [28], and Liu et al. [29] were among the first groups to pioneer the fabrication of insulin-loaded HA MNs. Liu et al. demonstrated that the novel insulin-loaded HA MNs exhibited self-dissolving properties and appeared safe. Hygroscopy, stability, drug release profiles, and dissolution characteristics of the insulin-loaded HA MNs were also characterized via a diabetic rat model by the same group [29].

Oral anesthetic injections without surface anesthesia can increase pain and anxiety during dental therapy. An HA MN patch with MN tips containing fast-dissolving lidocaine hydrochloride (LDC) was developed to address this problem. In the isolated porcine oral mucosa, LDC-containing HA MN patches were able to puncture the stratum corneum at a depth of approximately 0.28 mm. The fast-dissolving LDC on the HA MN patch enables anesthesia within 3 min at the tentative local anesthesia injection site. The adhesive MN patch was proposed to process the potential to aid transmucosal delivery of anesthetics, further enhancing the delivery of pain-free dental treatments. [30].

##### Sodium Alginate

Sodium alginate (SA) is a linear polysaccharide derived from brown algae. It is composed of poly-(1,4-diaminobenzoic acid) and nanoparticles of 1,4-L-glucuronide in different proportions, and its aqueous solution is highly viscous. As a result of the high number of carboxyl groups and hydroxyl groups in the molecular structure, SA has a high degree of chemical activity and can rapidly form a hydrogel containing a three-dimensional mesh structure, and it is non-toxic and odorless and exhibits outstanding biocompatibility and environmental friendliness. Safe dental application of alginate, e.g., dental impression materials, could be dated back to the 1950s [31]. SA and its derivatives are also readily available, inexpensive, simple, and renewable [32].

Blood loss is a common complication following trauma and surgery that can cause serious harm to the body [33]. Alginate is an excellent hemostatic polymer-based biomaterial as it is biocompatible, biodegradable, non-toxic, quickly gelled, and widely available [34]. In the medical field, SA hydrogels have been widely used, including injectable hydrogels [35], hemostatic needles [36], medical dressings [37], etc.

A North American group developed an injectable, biodegradable scaffold based on alginate microbeads for in vitro bone tissue engineering using periodontal ligament and gingival mesenchymal stem cells [38,39]. The stem cells remained viable in the laboratory and could differentiate into osteogenic and adipogenic tissues. Furthermore, the degradation behavior and swelling kinetics of the scaffold were characterized. Therefore, alginate has proven to be a promising non-toxic scaffold for stem cells, providing a good strategy for engineering bone tissue [40]. Additionally, a microneedle-mediated drug delivery system containing sodium alginate for immunochemotherapy has been developed. In glioma-bearing mice, microneedles loaded with lipopolysaccharide (LPS) and doxorubicin (DOX) demonstrated excellent efficacy in promoting immune response and inhibiting tumor growth [41].

##### Pullulan

Pullulan (PL) is an α-(1→6)-linked (1→4)-α-d-tri-glucosides polysaccharide/glucan or maltotriose, which is a carbohydrate biopolymer produced by *Aureobasidium pullulans*. The application of pullulan-based MN has been advocated since 2020 [42,43]. The authors present the first dissolving microneedle (DMN) system using PL. A variety of concentrations of PL gels were tested for viscosity and film formation, and then MNs were created using the appropriate concentration of PL gels. PL DMN was loaded with model molecules and proteins/peptides, and their stability was assessed using circular dichroism. Ex vivo studies were conducted using Franz diffusion cells to determine the permeation of Flu-Na and FITC-BSA-loaded PL-DMN into porcine skin. The findings suggest that PL DMNs can be effective for transdermal drug delivery [44].

In another study, transdermal insulin delivery was achieved using pullulan MN patches. MNs penetrated skin up to 0.38 mm depth and dissolved within two hours, releasing up to 87% of insulin. Storing insulin-loaded MNs at 4–40 °C for four weeks is possible without losing their structure. Additionally, PL MNs were non-cytotoxic, indicating they are suitable for skin application. A non-invasive treatment option for insulin could be made possible by PL MNs based on these findings [45].

##### Cellulose

In recent years, cellulose, a versatile and biocompatible material, has been garnering attention for its potential use in manufacturing microneedles for transdermal drug delivery [46]. Two studies have examined the use of cellulose-based microneedles and their unique properties. The one-step created semi-dissolving microneedles comprised a water-soluble needle layer and a backing layer containing 2,2,6,6-tetramethylpiperidine-1-oxyl-oxidized bacterial cellulose nanofibers. That material was used as drug reservoirs for delivering more significant quantities of drugs to the skin [47]. Another study examined the inclusion of cellulose nanofibers in dissolving microneedle arrays. Including cellulose nanofibers increased needle stiffness and decreased dissolving and transdermal delivery rates [48].

The cellulose-based microneedle is capable of delivering drugs transdermally. By leveraging the unique properties of cellulose, these microneedles improve drug loading capacity, enhance mechanical properties, and provide controlled drug release, which opens up new possibilities for advancing transdermal drug delivery systems in diverse applications, including oral disease management.

##### Combined Carbohydrate-Based Microneedles

Wei et al. [49] used CS and PL as raw materials to prepare HFMs. The chitosan-based MNs they developed showed good swelling and water retention properties and biocompatibility. The application of these hydrogel-based MNs as vehicles for drug delivery was then evaluated through a skin insertion study and a drug loading/release study. The mechanical properties of the MNs allowed easy insertion into newborn porcine skin with observed rapid release of drugs [49]. Combining these two polymers may produce synergistic effects, such as improved mechanical strength, swelling behavior, and drug loading capacity [50]. Due to the swelling and water retention properties of CS and the film-forming properties of pullulan, it may be possible to deliver and release medications more effectively. Moreover, the biocompatibility of these materials may reduce the possibility of adverse reactions or tissue damage.

#### 2.1.2. Protein-Based Microneedles

HFMs made from protein-based natural polymers are promising drug delivery systems that use biocompatible materials [51]. Proteins such as gelatin and silk fibroin are matrix-forming materials to manufacture MNs. They demonstrated significant potential in delivering transdermal drugs, vaccinations, and tissue engineering.

##### Water-Soluble Silk Fibroin

Silk fibroin (SF) is derived from silkworm silk after sericin degradation or degumming as a natural polymer. It comprises 18 amino acids forming a natural structural protein without physiological activity. SF is widely used in preparing sustained-release drug systems because of its biocompatibility, controllable degradation, and adjustable drug release [52,53]. The SF-based drug-sustained release systems can encapsulate and stabilize various small molecules, proteins, and large biomolecules such as DNA, facilitating controlled and prolonged content release [54]. In recent years, much attention has been given to using SF hydrogels in tissue engineering and drug delivery. It was reported that SF hydrogels prepared by various physical or chemical treatments, coupled with other biomaterials, provide a variety of drug release patterns, hence showing promise in the fields of cartilage tissue regeneration, wound repair, anti-cancer, and anti-infection therapies [55].

In an ectopic root canal transplantation model, silk fibroin scaffolds, not in MN format, though containing basic fibroblast growth factor (bFGF), were evaluated for pulp regeneration with DPSCs [56]. It was found that DPSCs seeded in the scaffold survived and displayed cytoplasmic elongation for at least four weeks in culture. Incorporating bFGF into tooth fragments and scaffolds increased the viability of DPSCs. This bFGF-incorporated scaffold generated pulp-like tissue consisting of transplanted and host-derived cells and displayed good vascularity, matrix deposition, and the formation of dentin-like tissue. The results of this study indicate that silk fibroin scaffolds incorporating bFGF are a promising candidate for future treatments in regenerative endodontics [57]. While some studies have explored the use of SF MNs for delivering other types of stem cells for tissue regeneration and wound healing applications, the specific combination of DPSCs and SF MNs in dentistry is yet to be explored.

##### Gelatin

Gelatin is a jelly-like substance derived from animals and composed of peptides and proteins released by partial hydrolysis of collagen, which are typically obtained from skin, bones, and connective tissues. The hydrolysis process breaks some bonds between and within the component proteins. Several gelatin chemical characteristics are similar to those of its parent collagen. In general, pig skin and cattle bones are used to manufacture photographic and pharmaceutical-grade gelatin. In terms of its composition, gelatin falls under the category of hydrogels. Various products use gelatin, including capsules, cosmetics, ointments, and foods. Gelatin MNs can be traced to as early as 2013 [58].

Gelatin has been shown to have osteogenic potential, which is often used as a tissue engineering scaffold for bone regeneration [59]. The efficacy of bFGF-gelatin hydrogel complex, not in MN presentation, was evaluated on bone regeneration around dental implants. A total of 24 titanium implants were placed into the mandibles of 4 beagle dogs, and different amounts of bFGF were applied to fill the bone defect sites. A minimum amount of bone regeneration was observed in the groups with 0 and 0.1 mg of bFGF after eight weeks, whereas new bone formation was observed in the groups with 1, 10, and 100 mg of bFGF and autogenous bone. The results suggest that bFGF-gelatin hydrogel complexes with an optimal amount of bFGF can be used to augment bone around implants [60]. Again, the specific combination of bFGF in gelatin hydrogel MNs aiding oral bone tissue healing is yet to be explored further.

#### 2.1.3. Mixed Carbohydrate-Protein Microneedles

One of the first gelatin MNs was a transdermal insulin delivery patch made from starch and gelatin developed by Ling and coworkers (2013). The MNs readily penetrate the test animals’ skin upon application and dissolve in five minutes. The effects of insulin-loaded MNs on diabetic rats were similar to those produced by subcutaneous injections, and the hormone-loaded MNs were stable after a month of storage [58].

In a study by Jana and coworkers [61], modified sodium carboxymethyl cellulose (CMC) and gelatin were used to fabricate DMN patches. The authors took advantage of the CMC-gelatin mechanical strength while the MN vehicle itself appeared able to prevent insulin degradation, thus potentially maintaining the shelf-life of this essential hormone.

### 2.2. Synthetic Polymer Fabricated Hydrogel-Forming Microneedles

Hydrogel MNs made from synthetic polymers can be designed to dissolve or swell in response to moisture or heat, allowing for the controlled release of drugs or other substances into the body. The mechanical properties of synthetic polymer fabricated MNs could be more substantial and usually more consistent than that of natural polymer fabricated MNs, thus improving their effectiveness and reliability. Additionally, synthetic polymers can be customized to acquire specific properties, including tunable degradation rates enabling more versatility for drug delivery.

#### 2.2.1. Gelatin Methacryloyl

Gelatin methacryloyl (GelMA) is a light-cure hydrogel developed by Van den Bulcke and coworkers [62]. The biocompatibility and controlled molding of GelMA led to the extensive application of the agent in the biomedical field shortly after its development and commercialization. By reacting methacrylic anhydride (MA) with amino groups, the double-bond amide groups would be grafted onto gelatin chains, then cross-linking of the latter would be facilitated through amide chemical reaction induced by UV activation of photoinitiators [63] (Figure 2). Since GelMA also exhibits good biodegradability and moldability, it has attracted considerable research interest. The unique characteristics of GelMA hydrogel and its simplicity of preparation made it a good MN candidate for wound healing dressings, drug delivery, biosensing, and tissue regeneration in a wide range of biomedical applications [64]. It was reported that GelMA could be used to prepare MN arrays with acceptable release profiles suitable for the delivery of water-soluble drugs [65]. GelMA hydrogel MNs have numerous applications in the medical field, such as transdermal insulin delivery, wound healing promotion, tissue regeneration, and biosensing. GelMA-based microneedles have been explored in wound healing, which possess adhesive properties and can be loaded with growth factors or other therapeutic agents or cells to promote wound healing processes. Incorporating GelMA microneedles into a wound site allows a controlled release of bioactive molecules, leading to accelerated tissue regeneration and improved healing outcomes [56,66,67]. Despite this, GelMA hydrogel MNs hold great potential for dental and oral applications due to their biocompatibility, biodegradability, and tunable mechanical properties, particularly local anesthesia, periodontal disease treatment, and/or controlled drug delivery.

#### 2.2.2. Methacrylate-Based Hyaluronic Acid

Methacrylate-based hyaluronic acid (HAMA) is obtained by chemically modifying the natural polysaccharide HA with MA [67]. This composite hydrogel has a continuous three-dimensional network structure, good swelling properties, mechanical properties, and drug loading capacity and is highly stable in a simulated human physiological microenvironment (pH 7.4) [68]. Xu et al. [69] developed a HAMA MN patch that carries platelet-derived growth factor D (PDGF-D) and human adipose-derived stem cells (ADSCs) for the management of diabetic ulcers. According to their diabetic mouse model, the MN patch delivered ADSCs and PDGF-D appeared to promote angiogenesis and wound healing.

At the dental front, a small molecule from the thiadiazolidinone family, NP928, was designed to be directly delivered into damaged teeth via a MAHA injection. Upon being cured by dental blue light securing in situ delivery, NP928 was observed to release from MAHA, which promoted the production of reparative dentine through upregulating Wnt/β-catenin activity in pulp stem cells [68]. Along such lines, the HAMA hydrogel-based MNs format may enhance another therapeutic agent’s delivery pathway in repairing damaged dental tissues.

#### 2.2.3. Polyvinyl Alcohol

Polyvinyl alcohol (PVA) is a water-soluble polar polymer with excellent biocompatibility, biodegradability, inherent, non-toxic, and mechanical properties. It can be chemically or physically cross-linked to form a hydrogel, and due to its wound-healing properties via transforming growth factor beta (TGF-β) upregulation [70], PVA hydrogels are also widely used as a matrix material for wound dressings [71].

Stratum corneum poses a formidable barrier to effectively delivering large and/or charged macromolecules such as small interfering RNA (siRNA). The latter would be helpful in skin disorders management [72]. Despite the effectiveness of intradermal siRNA injections, the procedure is painful. MN arrays may be an alternate and effective way for nucleic acid delivery, including siRNAs in a less painful manner. To enable penetration of the skin barrier, a loadable, PVA-based dissolvable protrusion array device (PAD) was developed. A PAD-mediated siRNA delivery effectively silenced a transgenic reporter mouse model in the skin [73].

### 2.3. Combined Natural and Synthetic Polymer-Fabricated Hydrogel-Forming Microneedles

A study reported that PVA and chitosan (PVA/CS) nanofiber scaffolds are excellent regenerative endodontics models [74]. Using these nanofibers and ciprofloxacin and IDR-1002, a multifunctional scaffold with anti-biofilm and anti-inflammatory properties was created. It was demonstrated that tooth fragments filled with these nanofibers produced pulp-like tissue in vivo. In dentistry, this type of scaffold, in MN/MN patch presentation, could aid regeneration beyond the pulp revascularization [75].

## 3. Fabrication of Hydrogel-Forming Microneedles

Hydrogel MNs can be prepared in a variety of ways, varying with the hydrogel material(s) and cross-linking mechanism [76,77] (Table 2). As mentioned in the previous section, the materials used for the preparation include natural polymers and synthetic polymers, which can be physically cross-linked, electrostatically interacted, or chemically cross-linked to form a hydrogel with the needle patch formation, as described in this section.

In detail, the fabrication of hydrogel microneedles involves an intricate process that includes mold fabrication, hydrogel formulation, casting, and post-fabrication treatments. Initially, a microneedle-shaped mold is meticulously crafted using advanced photolithography [78] or micro-molding techniques. Additionally, biocompatible polymers are meticulously blended with a suitable crosslinking agent to create a tailored hydrogel formulation. The hydrogel formulation is carefully poured or injected into the meticulously fabricated mold cavities without air bubbles by judiciously applying vacuum or centrifugation. Next, the hydrogel is crosslinked chemically or physically to ensure it is uniform and structurally robust. In addition to dehydration, sterilization and surface modifications to enhance drug loading and release properties are some post-fabrication treatments that are meticulously handled after fabrication to enhance mechanical strength [79].

Currently, very few oral/dental MNs are available on the market. Many methods have been explored to produce MNs for dental use, including micro-molding, casting, electrospinning, and 3D-printed hydrogel-filled MN arrays. In recent years, these methods have proven to be promising for developing MNs for dental applications, such as localized delivery of drugs and vaccines into the oral cavity. However, more research is needed to explore the full potential of MNs in dentistry.

### 3.1. Micro-Molding Method

Micro-molding hydrogel MNs preparation is one standard method applied [80]. Typically, polydimethylsiloxane (PDMS) casts are used, which are cast around a solid master template and then cured at 70 °C for two hours. A negative micro-mold can be produced from the master template for HFM fabrication. Due to the reusable nature of the micro-mold, multiple HFM arrays could be produced easily and quickly, which is particularly advantageous for optimizing parameters. A two-step fabrication is a common technique for fabricating MNs for drug delivery, with active ingredients concentrated at the MN tips (Figure 3).

### 3.2. Casting

Casting is another standard method of fabricating HFM. By using photolithography or a microfabrication technique, a master mold with the desired MN shape is created. Hydrogel material solution is then poured into the mold, typically composed of a mixture of water-soluble monomers/catalysts and the active ingredient(s). To enable solidification, the aqueous mixture is then allowed to crosslink, typically through UV exposure, heat, and/or a chemical initiator. The MNs can then be removed from the mold and stored until use [81].

The casting method enables the manufacture of MNs with complex shapes and dimensions that can be tailored to specific applications [82]. Additionally, this method allows for incorporating bioactive agents such as drugs, peptides, or growth factors into the hydrogel solution, which can be released upon MN insertion into the skin [81]. The casting method is relatively simple, cost-effective, and scalable, making it an attractive option for producing hydrogel MNs for various biomedical applications.

### 3.3. Electrospinning

Electrospinning involves drawing a biodegradable polymer solution, e.g., PVA, into a fine jet using a high-voltage electric field. The solvent evaporates as the jet travels toward a collector, and the polymer solidifies into a fibrous mat. HFM can be created by coating fibers with a hydrogel material, such as polyvinylpyrrolidone [83] or HA. Parameters for electrospinning, e.g., voltage, flow rate, and distance between the needles and the collector, can be controlled to produce MNs of various shapes and sizes [84].

MNs produced by electrospinning have a high aspect ratio, which means their length exceeds their width, allowing them to penetrate the skin more efficiently and effectively [85]. Moreover, the electrospun polymer, by its nature, is capable of releasing the hydrogel-loaded drugs or active agents in a controlled manner because of the ability of manufacturers to control the diameter and length of the MNs and porous structure by adjusting the electrospinning parameters [86]. However, electrospinning is among the more complex processes and requires specialized equipment.

### 3.4. Three-Dimensional-Printed Hydrogel-Filled Microneedle Array

It remains uncommon that existing hydrogel-filled MN arrays are designed to deliver different therapeutic agents to diverse tissue compartments at particular vicinity and to reach different depths of the targeted tissue mass. To fulfill this, Barnum and coworkers [87] developed an MN array system composed of a 3D-printed resin-based rigid outer layer attached to a drug-washed hydrogel. By fabricating MNs of varying compositions and lengths in a single patch, the same or different drug(s) can be delivered to various depths within the target tissue mass, perhaps with varying dosing protocols. The composition of the hydrogel and the shape of the needles can be adjusted in addition to the spatial distribution of the drug(s) involved. The delivery of vascular endothelial growth factor using a hydrogel-filled MN array was pioneered as a proof-of-concept approach [87].

Informed by various imaging techniques, the hydrogel-filled, 3D-printed, UV-polymerized resin-based, custom-made MN array of different needle densities, shapes, and lengths can be designed to acquire various forms and mechanical properties to ensure successful drug and biologics delivery. In line with the abovementioned concept, polyethylene glycol diacrylate hydrogel was developed, which retains properties for therapeutic drug delivery and skin penetration efficacy. In that study, an array of 100 MNs was successfully printed that release drugs in response to delivery site stimuli/characteristics such as temperature and pH [88].

## 4. Application of Hydrogel-Forming Microneedles in the Diagnosis and Treatment of Diseases

Hydrogel MNs are widely used in the diagnosis or treatment of diseases [89], i.e., extraction of skin interstitial fluid to detect health-relevant biomarkers in the human body [12], such as glucose, nucleic acid, proteins, etc., or they have been employed in tumors [90], diabetes [91], or eye infection management [92].

### 4.1. Diagnosis

Most research on HFM-mediated diagnosis has focused on medical applications, such as diagnosing skin diseases and various forms of cancer. As elaborated in the following section, HFMs may also be used in dentistry, for example, in diagnosing and treating oral conditions.

#### 4.1.1. Interstitial Fluid Extraction by Hydrogel-Forming Microneedles

Interstitial fluid (ISF) is a ubiquitous element within the parenchyma, enabling oxygen and nutrients exchange with waste products from cells. Biomarker analysis of ISF provides the opportunity for health evaluation of the tissue concerned [93]. In recent years, MNs extraction of ISF from the skin has received increasing attention in minimally invasive diagnostics and biosensing. Minimally invasive techniques are practical and patient-friendly methods aiding disease diagnosis such as superficial cancer, diabetes, and skin lesions, principally through ISF sampling, while minimizing the risk of blood contamination for detecting cell-free nucleic acid biomarkers [94]. As mentioned earlier, HFM materials with excellent swelling capacity would be the ideal agent for extracting ISF. Typically, immune cells are studied by analyzing blood cells, but this does not reflect their behavior in target tissue/organs. Mandal and coworkers developed an MN approach to sample tissue-resident immune cells in the skin. Utilizing this tool, they studied immune responses over time, improving the comprehension of how the immune system functions within the tissue tested [95].

Zheng and colleagues introduced an osmosis-driven hydrogel MN system consisting of osmolytes (maltose) [96] and showed up to 7.9/3.8 μL ISF could be extracted from isolated porcine skin/living mouse skin within 3 min, compared to 10 min when hydrogel MNs were used without osmolytes.

Al Sulaiman et al. [97] designed an SA-based hybrid hydrogel with linker DNA-cross-linked peptide nucleic acids (PNA/DNA). The setup is supposed to, within 2 min, detect specific/target nucleic acid biomarkers sequences, e.g., miRNAs in the target tissue ISF. With the MN array PNA/DNA platform/technology, fast sampling kinetics, relatively high sampling capacity, and especially multiple sites sampling could be achieved. The authors argued that the PNA/DNA MN array and an automated platform might enable longitudinal monitoring of local/systemic patient-specific health status in a personalized medical approach [97].

Upon satisfactory HFM-facilitated ISF extraction, the quantity of glucose or any drug substances within ISF could be measured. From blended hydrolyzed poly (methyl-vinyl ether-co-maleic anhydride) and poly (ethylene glycol) MNs applied on the skin of rat or human volunteers, concentrations of caffeine, theophylline, and glucose within extracted ISF could be measured by high-performance liquid chromatography with or without a proprietary kit. The analytes’ concentration showed a good correlation with their corresponding blood level [12].

#### 4.1.2. In Vivo Interstitial Fluid Real-Time Analysis

Alongside the development of MNs for the administration of diabetes medications, MN-integrated sensors for ISF glucose measurement [98] were also developed as an alternative to continuous glucose monitor devices, which enhanced patient compliance, excluding the application of lancets and the pain associated [99]. Zhao and coworkers [100,101] developed an MN biosensor using a filamentous protein/D-sorbitol array embedded with platinum and silver wires and glucose oxidase enzymes for continuous glucose monitoring. Glucose oxidase catalyzes glucose oxidation in the working electrode, producing hydrogen peroxide when inserted into the skin tissue. The working electrode then detects the hydrogen peroxide concentration, generating a current proportional to the initial glucose concentration [99]. In the future development of glucose-monitoring MN biosensors (c.f. Section 8.1), an array of microneedles loaded with a glucose oxidase enzyme and hypoglycemic drugs could be used for glucose detection and drug delivery. Microneedles may release medications in response to changes in glucose levels, depending on whether glucose levels are high or low. Consequently, this may enhance drug-dosing accuracy and reduce the risk of hypoglycemia and hyperglycemia. HFMs offer several advantages over traditional diagnostic methods, including biomarker detection, point-of-care testing, enhanced sensitivity and specificity, minimally invasive sampling, reduced risk of contamination, and multiplexed analysis.

#### 4.1.3. Epstein-Barr Virus Detection

Biomolecules of circulating extracellular vesicles, such as proteins and nucleic acids, can directly inform disease status in a living being [102]. Yang and coworkers [103] introduced a hydrogel MN patch for rapid 15 min ISF capture of Epstein-Barr virus (EBV) cell-free DNA. In vivo data indicated that EBV-DNA was released from nasopharyngeal carcinoma cells with a maximum capture efficiency of 93.6%. Based on that, the MN patch could have significant application potential [103].

### 4.2. Diseases Management

Various disease management applications have been demonstrated to be possible using HFM because of their ability to deliver drugs with minimal invasiveness and painlessly. They have been used to administer insulin to manage diabetes [91,104,105], along with vaccines and medicines to treat skin diseases such as psoriasis [106]. Additionally, hydrogel MNs have been used to deliver cancer drugs locally, and with antibiotics for treating bacterial infections [90]. They have also been explored for treating ocular infection [92], in which drugs can be delivered directly to the eye without an injection. In addition to their applications in drug delivery, HFMs have also shown potential in the field of microencapsulated cell delivery, opening new avenues for disease management [107].

#### 4.2.1. Cancer

Surgical tumor resection is extremely technique-sensitive and invariably associated with various levels of morbidity. If malignant tumor resection is incomplete or total resection is not possible, it results in a high risk of tumor recurrence. Chemotherapy and/or irradiation therapy, post-surgical intervention or not, could be highly stressful and is often associated with many cases of collateral damage, side effects, and suffering. Oral and parental routes of anticancer drug administration are always associated with systemic adverse effects and, nevertheless, remain at risk of eventual low anticancer drug bioavailability at the site of action because of pharmaceutical limitations and physiological barriers. As a new generation of trans-tissue local drug delivery [108], hydrogel MNs loaded with anticancer drug(s), if appliable in the vicinity of the primary tumor resection site, with or without the aid of endoscopy, can secure effective drug delivery to ensure tumor elimination and/or minimize tumor recurrence [109] as well as reduced systemic adverse effects [110].

Hydrogel MNs can be used for transdermal delivery of small molecules to improve the bioavailability of drugs. Huang and colleagues [90] synthesized photo-crosslinked dextran methylmalonic acid as the carrier of a novel MN with DOX and trametinib (Tra)-loaded tips, which can penetrate the epidermis to achieve sustained release of drugs. Tra can reverse P-glycoprotein-mediated multidrug resistance and effectively block the efflux of P-glycoprotein to DOX. The results showed a synergistic effect of simultaneous release of Tra and DOX in a B16 cell xenograft nude mouse model [90].

Cancer immunotherapy involves the provocation of the body’s immune system to fight against cancer cells (Figure 4). Several studies were conducted on using HFM to deliver immunotherapy in cancer treatment [111]. Kim and coworkers [112] examined the use of hydrogel MNs to deliver checkpoint inhibitors, a type of immunotherapy drug used in cancer treatment. Checkpoint inhibitors block specific proteins in the body that prevent the immune system from attacking cancer cells. Photolithographic-loaded checkpoint inhibitor-hydrogel MNs were developed and applied to a cancer animal model, confirming the effectiveness of the MN protocol in enhancing the immune response against cancer cells and reducing tumor growth [112]. Microencapsulation techniques have emerged as effective strategies in cell therapy, providing immune-privileged environments, mechanical support, and controlled release capabilities through MN array patches, thereby improving long-term cell survival and treatment outcomes [113].

#### 4.2.2. Diabetes

Diabetes is a common chronic metabolic disease, and blood glucose levels are often difficult to control in those severely affected. High compliance is needed because patients must monitor their own blood glucose on a daily basis (c.f. Section 4.1.1 and Section 4.1.2) while taking their medications and/or regular insulin injections, if needed, on top of a healthy lifestyle. Hydrogel MN provides controlled insulin release.

Chen and coworkers devised a prolonged hydrogel MN insulin release protocol using bio-compatible SF protein and phenylboronic acid/acrylamide, with SF as the MN base layer and glucose-responsive hydrogel forming the tip layer. This hydrogel micro-needle remains in its original shape after a week in the aqueous system, hence providing an opportunity for continuous drug release triggered by the addition of glucose oxidase. When the glucose concentration rises, it leads to a decrease in the pH inside the MNs, an increase in the pore size of the MNs, and an acceleration in the rate of insulin release. [91]. A ‘smart insulin delivery’ protocol to regulate blood glucose levels was designed by Wang and colleagues using PVA MN with glucose oxygenase as a core and peroxisomal enzyme embedded in the outer shell, which is responsive to elevated blood glucose concentrations. To enable oxidative distress or H_2_O_2_-triggered insulin release, insulin was chemically modified with 4-nitrophenyl 4-(4,4,5,5-tetramethyl-1,3,2-dioxol-2yl) benzyl carbonate (insulin NBC) and then loaded onto a water-soluble PVA substrate. When the MN is exposed to high glucose concentration, high localized levels of H_2_O_2_ are generated, and insulin NBC is oxidized and hydrolyzed, leading to the rapid release of free insulin and therefore aiding control of blood glucose concentration (Figure 5). This core-shell structured MN provides a painless, ‘on-demand’ technique for transdermal drug delivery in a primarily physiologically factor-controlled manner, improving bio-compatibility [104].

#### 4.2.3. Rheumatoid Arthritis

Rheumatoid arthritis (RA) is an inflammatory autoimmune disease mainly affecting joints. The condition has a complex etiology, and those affected exhibit complicated pathogenesis and multiple joint involvements [114]. Local administration of drugs preventing/attenuating joint deformities in the clinical setting is complicated, e.g., local drug injection poses risks of joint damage and infection. Hydrogel MNs can be used as a vehicle for drug and protein inhibitor delivery to the joints involved, minimizing the adverse/side effects of RA drugs on the gastrointestinal tract if administered orally [115].

Cao and colleagues [116] studied a chemically modified aptamer delivered by translucent acid hydrogel MN to treat RA. The authors first modified DTA (DEK-targeted aptamers, a nucleic acid aptamer that blocks the ubiquitous chromatin DEK proteins) using methoxy and reverse deoxythymidine DTA4 on the ribose unit and then modified DTA4 with cholesterol at its 5′ end to form DTA6, which was loaded onto a methacrylate-modified HA hydrogel MN that rapidly released DTA6 into the dermis upon skin puncture. The in vitro results showed that DTA6 significantly reduced DEK expression in inflammatory RAW264.7 cells, while in vivo it protected mouse joints from cartilage/bone erosion [116].

#### 4.2.4. Psoriasis

Psoriasis treatment with methotrexate (MTX) is limited by its side effects [117]. It is desirable to deliver drugs transdermally, but currently, available techniques have limitations. Researchers have developed a dissolving MN patch made of HA with controlled doses of MTX to overcome these limitations. The MNs successfully delivered MTX intralesionally and exhibited anti-inflammatory effects in mice with psoriasis-like skin inflammation, with higher efficacy than oral administration [106]. Recently, Liu’s team developed a new type of MN patch, which is H_2_O_2_ responsive. It contains quickly released MTX and epigallocatechin gallate, which has a sustained release in reaction with H2O2. It showed improved outcomes in psoriasis therapy [118].

#### 4.2.5. Eye Infection

A Hong Kong/Singaporean research team developed CryoMN patches for delivering *Bdellovibrio bacteriovorus*, a predatory bacterium preying on other gram-negative bacteria, to treat ocular infections. These MNs kept 80% viability of *B. bacteriovorus* in long-term storage and showed gram-negative bacteria inhibition in vitro. In a porcine model, the agent reduces infection by almost 6 times over 2.5 days without effects on cornea thickness or morphology [92].

#### 4.2.6. Microencapsulation Cell Delivery

Microencapsulation of cells within HFMs provides a protective environment for implanted cells, allowing for prolonged cell survival and therapeutic efficacy. This approach has been explored in various disease management strategies, including vaccines [119], tissue engineering, and cell-based therapies [120]. By encapsulating cells within HFMs, controlled release of cells and their therapeutic factors can be achieved, enabling targeted delivery and enhancing treatment outcomes. A research group developed a novel approach for treating vulvovaginal candidiasis, an infection caused by *Candida* sp., by formulating *Lactobacillus plantarum*-loaded microcapsules using HFMs [121].

## 5. Applications of Hydrogel-Forming Microneedles in Dental Therapy

Over the past few years, researchers and clinicians embracing HFM applications have actively explored applying such technology and/or novel concepts to improve oral health care and prevention. The following is an overview of laboratory-based and clinical studies regarding recent explorations and/or attempts to translate HFM concepts for oral condition/disease prevention, diagnosis, treatment, and beyond (Figure 6).

### 5.1. Collecting Oral Fluid for Diagnosis

In the past, gingival crevicular fluids (GCF) were collected through paper strips [122]. The use of this method, however, presents several challenges, including contamination of GCF samples by saliva, plaque, and blood, while only small amounts of GCF are being collected [123]. Moreover, retrieving all GCF content from paper strips has proven to be difficult. These challenges, however, may be overcome by using MNs. An HFM patch can be used in dentistry to collect gingival/periodontal ISF. In addition to identifying and quantifying various biomarkers, including the possibility of continuous monitoring of periodontal health/diseases, these could also serve as prognostic indicators for the outcome of periodontal therapy. In the current state of research, and due to limited studies, it is difficult to determine the effectiveness of HFM in collecting oral fluids, such as gingival ISF. Past reports have focused on MN-collected ISF from the skin, particularly in experimental animals, while recent studies reported that HFMs could be applied to collect saliva, blood, and other types of oral fluids, which can be used for diagnostic and therapeutic purposes.

### 5.2. Early Detection of Oral Cancer

The asymptomatic nature of oral cancers, such as oral squamous cell carcinoma, makes it difficult to be detected at the early stage. Several biomarkers, including lactic acid and valine, tissue polypeptide antigen, lysine, proline, citrulline, and ornithine, have been advocated to be associated with oral cancer [124]. Early detection of these biomarkers in ISF from suspected oral lesions collected using MN-based devices would be a good way forward.

### 5.3. Oral Cancer Treatment

Since the filing of MN patents in the 1970s, significant progress has been observed, especially in using MNs as a drug delivery system. In recent years, MNs have also been used to deliver drugs across the oral mucosal barrier. The treatment of recurrent oral cancer is challenging because surgery plus/minus chemotherapy and/or irradiation therapy may not achieve the desired therapeutic effects due to already massive loss of maxillofacial structures, scarring, and insufficient perfusions due to the prior treatments at tumor first detection. Local intra-tumor injection of anti-cancer drugs is an alternative therapeutic approach. However, due to the sequelae of previous therapy, this method leads to uneven distribution of drugs in the tumor and causes pain and suffering to the patients concerned. Poly(lactic-co-glycolic acid) (PLGA) coated with DOX with an average particle size of 137 nm was prepared to address the above challenges [125]. On the MN array, DOX-PLGA nanoparticles coated with a precalculated, sufficient dose of cytotoxic DOX to cancer cells located within a 1 cm radius from the MN array enabled cell death in a 3D tumor tissue model. Confocal microscopy and stereomicroscope also demonstrated that DOX could diffuse into porcine oral tissues deep and laterally over an area of 1 × 1 cm^2^ [125]. Another research group developed a cellulose MN patch for localized chemotherapeutic agent 5-fluorouracil delivery to the buccal mucosa to treat oral carcinoma [126]. A photosensitizer-loaded microneedle patch was developed for targeted phototherapy in oral carcinoma. The patch successfully penetrated the buccal mucosa and dissolved within a short time. Localized application of the patch, followed by near-infrared laser irradiation, significantly reduced tumor volume in a xenograft tumor model [127].

### 5.4. Oral Vaccination

Compared to the gastrointestinal environment, the oral milieu provides a less harsh situation for mucosal vaccination, thus preventing significant degradation of vaccine antigens [128]. The presence of IgA-mediated immunity, in particular, enables the oral cavity to be one of the prime sites for vaccinations against infectious agents [129].

The effectiveness of oral mucosal vaccination depends on the permeability of the mucosa, which is influenced by the thickness of the epithelium and the degree of keratinization. The sublingual mucosa is more permeable than the oral and gingival mucosa, so sublingual vaccination has been extensively investigated [130]. Ma and coworkers reported that coated MN used in oral cavity vaccine delivery activated systemic and mucosal immunity, while intramuscular injection can only activate systemic immunity [131]. They soon realized the inferior stability and limited dose capacity of the solid or coated MNs. Zhen and colleagues [132] used hydrogel MNs containing mannose-polyethylene glycol-cholesterol/lipid A-liposomes (MLLs) to encapsulate the target antigen, establishing a proMLLs-filled MN array. Antigens with adjuvant loaded into MNs were administered to mice via the oral mucosa route, and the entire immunization course resulted in systemic and mucosal immune responses within three weeks: mixed Th1/Th2 immune response and enhanced levels of IgG2a, IFN-γ, and CD8+ T cells. [132].

### 5.5. Local Anesthesia

The general population has a 1.6% prevalence of dental injection phobia [133]. To improve patient compliance during dental, periodontal, surgical, and endodontic procedures, topical anesthesia delivered by MNs might help the management of injection phobia. HFM technology has been applied in dentistry to enhance surface anesthesia. Daly and colleagues [134] conducted a randomized controlled trial of 0.5 g 5% lidocaine gel (Septodon^®^) surface anesthetics directly delivered with MN patches to relieve dental injection pain in fifteen participants randomly divided into four groups with a quasiexperimental-crossover design. The test sites (buccal or palatal mucosa) received ‘anesthetic gel applied onto the center of an MN patch’ delivery, while the control sites received the anesthetic gel directly. Pain related to the needle penetrated the oral mucosa only, the needle inserted through the oral mucosa to contact bone, and the needle inserted through the oral mucosa and an entire cartridge of local anesthetic administration were evaluated using visual analog scoring (VAS) and a four-point verbal rating scale (VRS). According to the authors, the MN patch plus lidocaine gel significantly reduced VAS pain scores at both buccal and palatal sites, and VRS pain scores were also significantly reduced compared to the control sites [134].

On the other hand, using conductive MN arrays, Seeni et al. applied low-voltage iontophoresis to target dental sensory nerves, creating micro-conduits that accelerate local anesthetic drugs to block the sensory supply in a rabbit model [135].

### 5.6. Caries Prevention

Cariogenic oral bacteria produce acids from dietary carbohydrates, leading to enamel and dentine demineralization, hence dental caries [136]. Calcium and phosphorus ions in saliva can reduce or reverse this condition [137]. Fluoride is another mineral that can prevent demineralization [138]. The presence of calcium, phosphorus, and fluoride in saliva reduces the activity of these bacteria and facilitates the process of remineralization. These minerals should be present continuously to promote remineralization. However, because of poor diet or hyposalivation, such minerals might not be present sufficiently and at the proper pH to protect the host from dental caries [139].

MNs providing a slow release of fluoride or other remineralizing agents for caries prevention were experimented upon. One approach is to target the leading cause of dental caries, i.e., *Streptococcus mutans*. Many metal and metal oxide nanoparticles (e.g., nano-silver, nano-hydrous zinc oxide, nano-gold) were shown to destroy this caries pathogen even in the deeper layers of the biofilm [140]. MNs or nanosheets mediated metal and metal oxide nanoparticles delivery could reduce the risk of dental caries. The current attempt was to apply an anti-caries agent-loaded HFM patch inserted into oral mucosa juxtaposed to its anticipated site of action. Previous studies have shown that perhaps even the slightly increased fluoride content and remineralizing agents in gingival crevicular fluid and saliva could enable reductions in tooth decay [141,142].

### 5.7. Oral Ulcer Management

An oral ulcer is a common locally acute and painful oral inflammatory disease. A fraction of the world’s population suffers from recurrent or chronic oral mucosal lesions that can negatively impact one’s oral health-related quality of life, and in rare cases, it can be the precursor of oral cancer [143]. It has been proposed that soluble HA MN patches containing betamethasone 21-phosphate sodium and betamethasone 17,21-dipropionate might be used. HA loaded with the medication applied in a rat tongue mucosa ulcer model successfully secured the release of the drugs into the ulcer base with facilitated wound healing on top of anti-inflammation, making the approach a viable treatment option for humans [144]. Another HA-based microneedle patch loaded with multi-drugs, including dexamethasone acetate, vitamin C, and tetracaine hydrochloride, has been developed to treat recurrent oral ulcers. The patch exhibits enhanced anti-inflammatory and pro-proliferation effects compared to individual components for recurrent ulcer treatment [145].

### 5.8. Periodontal Treatment

Periodontitis, a chronic oral inflammatory disease, is one significant oral health burden worldwide. The development of this disease is due to dysbiotic subgingival biofilms, which induce dysregulated immuno-inflammatory responses in a susceptible host [146]. A modular MN patch of gelatin and PLGA-carrying tetracycline and IL-4 and TGF-β-loaded silica nanoparticles was developed by Zhang and coworkers [147] to deliver immunomodulation to gingival tissue to facilitate tissue regeneration [147]. Application of the MN patch to rat or mouse periodontitis models inhibited inflammatory factors and encouraged regeneration-promoting signals and tissue healing [147].

## 6. Advantages of Hydrogel-Forming Microneedles in Dentistry

As MN biomedicine research has advanced over the past decade, the advantages of MN systems have become apparent: (1) drug delivery with minimal discomfort: HFMs penetrate the oral mucosa without causing significant ache, making them ideal for pain-free drug delivery in dentistry; (2) targeted drug delivery: the nature of local HFM application means drugs directly delivered to the site(s)/vicinity perhaps at predetermined tissue depths where they are needed, i.e., through oral mucosa to targeted maxillofacial locations without the necessity of the drug first going through systemic circulation or liver first-pass metabolism; (3) controlled drug release: HFMs may be designed to release drugs at a controlled rate, resulting in locally high dose and prolonged drug delivery; and (4) improved patient compliance: for home-administrated medications, the controlled local HFM delivery usually extends the dosing intervals to be hopefully more convenient to patients, hence enhancing compliance.

Compared with traditional metal- and silicon-based MNs, hydrogel-forming MNs exhibit the following advantages: (1) fair drug carriage capacity with controlled drug release; (2) the polymeric biomaterials needle body is biocompatible and avoids foreign body reactions when MN tips fracture.

The therapeutic approach of microneedles has gained widespread recognition among clinicians of various organizations, as it is considered a clinical treatment or drug delivery modality with high potential. The transdermal drug delivery method of microneedles is generally believed to possess several advantages over traditional oral or injectable modes of administration: (1) The drug delivery method of microneedles can bypass the acidic environment of the gastrointestinal tract or the action of digestive enzymes, thereby potentially increasing the drug’s bioavailability and enhancing its therapeutic efficacy to some extent; (2) compared to the injection route, the drug delivery method of microneedles is a painless and non-invasive approach. The use of nano- or micro-meter-scale microneedles reduces patient discomfort during skin penetration. Additionally, this delivery method minimizes tissue damage and improves treatment safety; (3) microneedle administration is simple and convenient, eliminating patient discomfort associated with swallowing pills or receiving injections, which in turn, can improve patient compliance; (4) the size and shape of microneedles can be individually customized to accommodate different targeted populations and applications, catering to personalized drug delivery modes in various environments and processes; (5) microneedle arrays offer the possibility of controlled drug release, allowing for reduced dosing frequency and aid maintenance of adequate and stable medication therapeutic concentration/level in a patient’s bloodstream. Therefore, applying microneedles as a drug delivery modality represents a promising approach.

## 7. Limitations of Hydrogel-Forming Microneedles in Dentistry

Only a few clinically marketed oral HFM applications are available, even though preclinical studies have shown promise related to this novel interventional approach. Additionally, a few shortcomings remain to be addressed: (1) MNs, as the name implied, were of limited capacity, so they could only deliver a small load of drug(s) before changing for a new HFM patch needed; (2) limited clinical evidence regarding HFMs’ efficacies in dental application: as described in this narrative review, HFMs rely upon trans-mucosal drug delivery. Therefore, can HFM devices be engineered for direct caries prevention and management? (3) The HFM-mediated therapeutic results would be affected by the mucosal conditions of the ‘MNs bed’, and the nature of MN base material used; (4) commercialization or scale-ups of MN patches production perhaps is yet to be realized; (5) HFMs may have limited needle strength, which can lead to bending or deformation during insertion and reduce their effectiveness in penetrating the keratinized or non-keratinized mucosa/skin and delivering drugs to the desired depth; (6) HFMs may exhibit variable drug release kinetics, leading to inconsistent drug delivery profiles. Achieving precise control over drug release rates and durations can be difficult; (7) storage and stability concerns: HFMs require proper storage conditions to maintain their integrity and stability. Factors such as temperature, humidity, and exposure to light can affect their performance and shelf life.

Importantly, the extent and significance of these disadvantages may vary depending on the specific design, formulation, and application of hydrogel-forming microneedles. To address these challenges and improve the performance and practicality of HFMs for a range of biomedical applications, continuous research and development efforts are being undertaken.

## 8. Future Perspectives

The use of hydrogel-forming microneedles in medical research, especially concerning targeted or local delivery of drugs, has shown great promise. Nevertheless, the use of hydrogel MNs in dental applications remains relatively limited, and more research is required to explore all potentials of the novel protocol. Compared with other types of MNs, HFMs have received considerable attention and study due to their simple preparation method and good biocompatibility, making them the most promising MNs for clinical applications [148]. Exploring the use of new or modifications of existing materials can improve the stability and performance of HFMs.

### 8.1. Hydrogel-Forming Microneedle Biosensing

MN-based biosensors are anticipated to become an important part of future medical services as research advances. The development of integrated diagnostic and therapeutic MNs, which combine the transdermal drug delivery capability of MNs, will allow patients with chronic diseases to undergo frequent testing and receive medication. The data transmission module can be integrated into the system so that patient tests can be uploaded, and doctors may treat and diagnose patients online based on the data recorded, thereby reducing the dependence on medical personnel. Additionally, MN biosensors allow patients to self-regulate based on their results. The devices could potentially be used as diagnostic tools for studying oral fluids and for remote monitoring of oral health. It is also possible to use HFMs for the targeted delivery of contrast agents in dentistry to enhance imaging. Consequently, diagnostic accuracy and treatment planning may be improved. Using 3D-printing technology to design MNs according to the needs of patients, future research directions and development should focus on reflecting personalized characteristics, thereby ensuring that dental treatment is more effective and safer.

### 8.2. Accelerate Dentoalveolar/Periodontal Wound Healing

In theory, microneedling is perhaps an adjunct to dentoalveolar/periodontal healing because it could speed up wound healing. In dermatology, microneedling can cause superficial bleeding and trigger a wound-healing cascade via the release of various growth factors [149]. Hydrogel serves as a perfect tool to treat wounds [150]. As fibroblasts migrate, proliferate, and lay down the intercellular matrix, neovascularization, and collagen formation are promoted [151]. Conventional wound closure methods may result in tissue damage and inflammation, especially in the oral cavity. A double-layered adhesive MN patch made from mussel adhesive protein and SF was developed to promote better wound healing. Patches can be inserted into tissue with a 7× greater force than skin penetration, swell to seal luminal leaks, and exhibit excellent wound healing in vivo [13]. There are many practical applications for this bioinspired adhesive patch, including wound healing and delivering regenerative or anti-inflammatory agents transdermally.

Green tea (GT) extract contains catechins that disrupt the cell membranes of bacteria, inhibiting gram-positive bacteria [152]. GT extract and hyaluronic acid can be used as antibacterial gingival microneedle patches to promote the healing of periodontal wounds, which have good slow-release properties, are not toxic to normal cells, and inhibit bacterial growth even at low concentrations. According to a recently published study, experiments on oral wounds of diabetic rats showed that hydrogel adhesive could effectively protect against mucosal wounds and shorten the inflammatory phase, thus promoting the wound-healing process [153]. In rat models, microneedles have been found to accelerate wound healing and inhibit bacterial growth [154]. In future research, HFM can be designed to release growth factors, cytokines, and other molecules that promote periodontal tissue regeneration and wound healing. By delivering these molecules directly to the site of the dental injury or surgery, hydrogel MNs can accelerate healing and reduce recovery time.

### 8.3. Anti-Bacterial Effects

The application of novel MN patches, which are safe and self-administrable and loaded with anti-bacterial agents, can maintain an optimal level of active compound in the oral cavity for a more extended period of time, thus increasing their effectiveness [155]. Nano-silver has proven to display adjuvant immune properties and antimicrobial properties. Therefore, incorporating nano-silver into MN patches may provide dual benefits [156]. CMC micropatches with nano-silver enhance antimicrobial activity against gram-positive and gram-negative microbes. Through the use of self-DMN that are infused with gelatin nanoparticles (GNPs), antibiotics can be delivered to active sites within biofilms [157]. Chloramphenicol@GNPs are more effective against *Vibrio vulnificus* biofilms than direct administration of CAM. Biofilm-contaminated sites can now be treated more effectively with this new treatment strategy [158]. Another study investigated the use of HFM for transdermal drug delivery of antibiotics, demonstrating controlled release properties and potent antimicrobial activity against *Escherichia coli* and *Staphylococcus aureus*. The microneedles were fabricated using a custom design 3D-printed master template to improve resolution and successful penetration of porcine skin grafts [159]. In the future, there is a potential to develop hydrogel MNs that can target specific bacterial strains in the oral cavity, such as *S. mutans*, which is a major contributor to dental caries. Additionally, hydrogel MNs can provide a sustained release of antimicrobial agents, reducing the need for repeated administration and improving patient compliance. Hydrogel MNs can also be designed to release agents in response to specific stimuli, such as changes in pH or the presence of bacterial enzymes, leading to targeted and efficient treatment of oral infections.

### 8.4. Future Research to Evaluate the Efficacy and Safety of HFMs in Dental Applications

Future research could further evaluate the efficacy and safety of hydrogel microneedles in dental applications. This could involve conducting in vivo studies to assess their performance in various dental treatments, such as periodontal disease management, local anesthesia delivery, or targeted drug delivery to oral tissues. Additionally, investigating the long-term biocompatibility and biodegradability of hydrogel microneedles in the oral cavity is crucial. Furthermore, comparative studies comparing the effectiveness of hydrogel microneedles with traditional treatment modalities will provide valuable insights. Finally, exploring the potential of functionalized hydrogel microneedles for diagnostic purposes and biosensing in dentistry should be considered. Overall, these future research directions will contribute to expanding the knowledge and utilization of hydrogel microneedles in dental applications.

## 9. Conclusions

The application of hydrogel-forming microneedles shows excellent potential for various oral diagnosis and dental disease management applications. Hydrogel-forming microneedles can collect oral fluids, deliver drugs, promote tissue regeneration, and release active substances over an extended period. Preclinical studies have indicated that hydrogel-forming microneedles may revolutionize how dentists diagnose and treat dental diseases, resulting in improved patient outcomes and reduced treatment time. However, further research and clinical studies are needed to fully evaluate the efficacy and safety of hydrogel-forming microneedles in dental applications.

## Figures and Tables

**Figure 1 materials-16-04805-f001:**
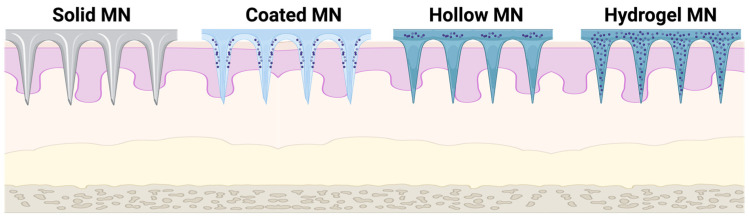
Microneedle (MN) types for transdermal drug delivery. Left to right: Solid MN—solid shaft with sharp tips for skin penetration and drug delivery by dissolution, diffusion, and surface coating; Coated MN—solid base with surface coating for controlled release of encapsulated drug/active ingredients; Hollow MN—needles with a hollow core for fluid injection or collection; Hydrogel MN—biocompatible needles that dissolve upon insertion, releasing loaded drugs or vaccines. The figure was created with BioRender.com.

**Figure 2 materials-16-04805-f002:**
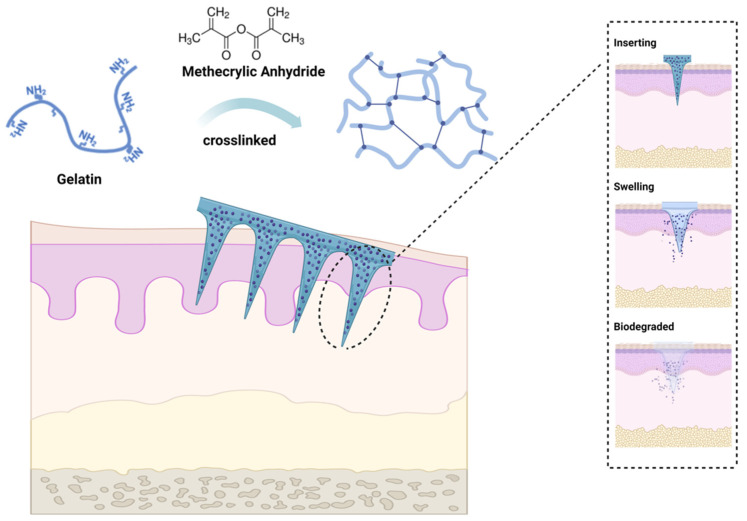
Gelatin methacryloyl (GelMA) microneedle application. GelMA is a cross-linked hydrogel material prepared by grafting methacrylic anhydride (MA) onto gelatin. GelMA MN exhibits sufficient mechanical toughness to penetrate the skin upon insertion, followed by swelling, degradation, and drug release, leaving no residue within the skin tissue. The figure was created with BioRender.com.

**Figure 3 materials-16-04805-f003:**
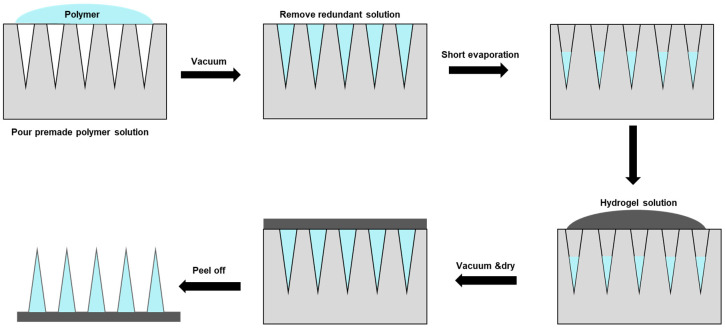
Hydrogel microneedle patch fabrication through a two-step method. First, the polymer solution with active ingredients was covered and filled in the polydimethylsiloxane (PDMS) female mold under a vacuum. Next, the redundant solution was removed to ensure each cavity was filled with the same volume of drug solution. The solution was then let to evaporate. Hydrogel solution was then poured into the mold under a vacuum, and the sample was allowed to cure/solidify and dry in a sealed desiccator overnight, followed by demolding, packaging, and storing.

**Figure 4 materials-16-04805-f004:**
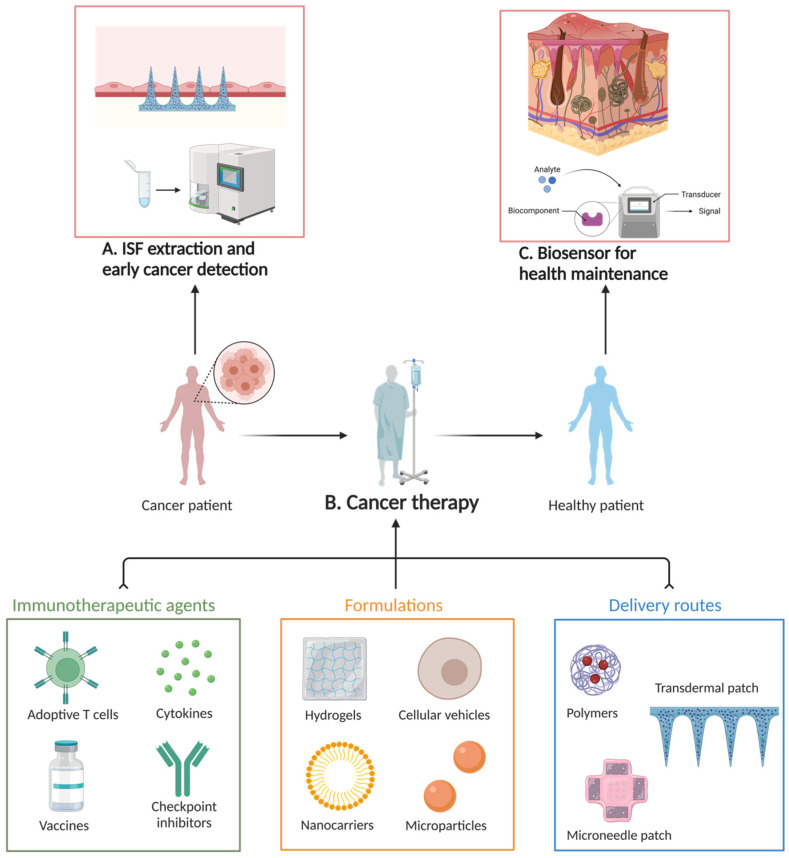
Schematic illustration of the potential applications of microneedles in cancer detection and management. (A) Microneedle device is used for interstitial fluid (ISF) extraction, which can be used for early cancer detection. ISF contains biomarkers that can indicate the presence of cancer in the body, making it a promising alternative to traditional blood tests for cancer screening. (B) Hydrogel microneedles can be used for the local delivery of cancer therapeutics. These microneedles can be loaded with adoptive T cells via microencapsulation, cytokines, vaccines, and checkpoint inhibitor drugs that target specific cancer cells or regions of the body, allowing for precise and targeted treatment. (C) Microneedles can be used as biosensors to monitor the health of cancer patients during and after treatment. The figure was created with BioRender.com.

**Figure 5 materials-16-04805-f005:**
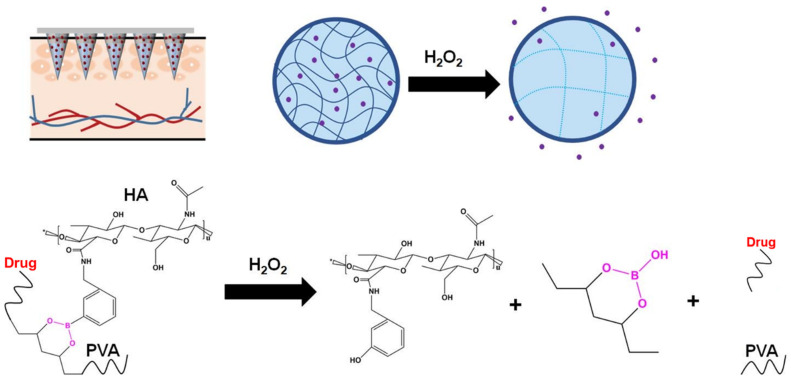
Schematic representation of the mechanism of H_2_O_2_-sensitive drug (red) release using 4-nitrophenyl 4-(4,4,5,5-tetramethyl-1,3,2-dioxol-2yl) benzyl carbonate (NBC; pink; drug-vehicle complex: purple dots) conjugated polyvinyl alcohol (PVA) substrate in hyaluronic acid (HA) microneedle (MN). NBC conjugated onto a water-soluble PVA substrate in MN format (upper left) is exposed to high glucose concentration, which generates high localized levels of H_2_O_2_. The H_2_O_2_ oxidizes and hydrolyzes NBC (upper center: pictorial; lower panel: line formula representation), rapidly releasing NBC and drugs from the HA MNs.

**Figure 6 materials-16-04805-f006:**
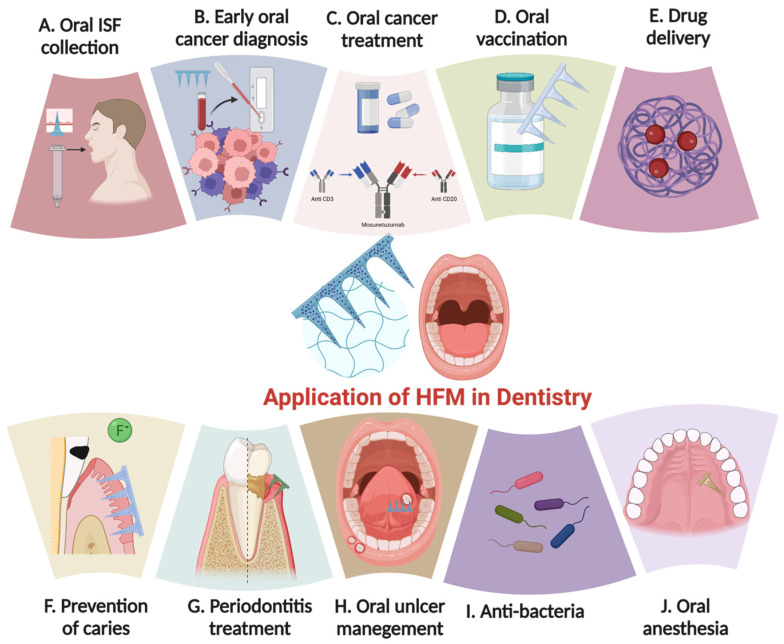
Hydrogel-forming microneedles (HFM) application concepts in dentistry. **The top panel from left to right** depicts the application of hydrogel microneedles (MNs) in the maintenance and/or promotion of systemic health: (**A**) Oral interstitial fluid collection, where the MNs are used to painlessly collect oral fluids for diagnostic purposes. (**B**) In early oral cancer diagnosis, the MNs can collect and analyze oral tissues and fluids for cancerous cells or related activities. (**C**) Oral cancer treatment, where the MNs can deliver precise doses of chemotherapy drugs directly to affected areas. (**D**) Oral vaccination, where the MNs can be used to deliver vaccines directly to oral mucosal tissues for more effective mucosal immunization. (**E**) Drug delivery, where the MNs can be used to deliver a wide range of drugs directly through oral tissues for improved systemic or local therapeutic outcomes. **Lower panel from left to right**: (**F**) Prevention of caries, where the MNs can deliver antibacterial agents to oral mucosa surfaces next to carious lesions to remineralize initial reversible tooth decay. (**G**) Periodontal therapy, where the MNs can deliver drugs directly to the gums to the adjunct periodontal treatment of gum diseases. (**H**) Oral ulcer management, where the MNs can deliver analgesic/anti-inflammatory drugs directly to the affected areas. (**I**) Antibacterial, where the MNs can deliver antibiotics directly to oral tissues to fight bacterial infections. (**J**) Oral anesthesia, where the MNs can provide localized pain relief for simple dental procedures or anesthetize the mucosa before local anesthetic injection. The figure was created with BioRender.com.

**Table 1 materials-16-04805-t001:** Materials used for hydrogel-forming microneedles.

Material	Description	Applications
*Natural Polymers*		
Carbohydrate-based	Polysaccharides with biocompatibility and bioactive properties	- Chitosan or chitosan derivatives: Biocompatible, biodegradable, antimicrobial properties. Used in transmucosal vaccine delivery and oral anesthetic patches. - Hyaluronic acid: Highly biocompatible, water-retention ability. Used in transdermal drug delivery and pain-free dental treatments. - Sodium alginate: Biocompatible, non-toxic, widely available. Used in hemostatic needles, medical dressings, and bone tissue engineering. - Pullulan: Dissolving microneedles with potential for transdermal drug delivery. - Combined carbohydrate-based microneedles: Synergistic effects of chitosan and pullulan, improved mechanical strength and drug loading capacity.
Protein-based	Gelatin and silk fibroin with biocompatibility and controllable degradation	- Water-soluble silk fibroin: Biocompatible, adjustable drug release. Used in sustained-release drug systems and tissue engineering. - Gelatin: Derived from collagen, used in capsules, ointments, and tissue engineering for bone regeneration.
*Synthetic Polymers*		
Polyvinyl pyrrolidone	Synthetic polymer with good mechanical properties and biocompatibility	- Used in drug delivery systems, such as insulin-loaded microneedles.
Poly (ethylene glycol)	Synthetic polymer with high water solubility and biocompatibility	- Used in hydrogel-forming microneedles for drug delivery.
Poly (methyl vinyl ether-co-maleic acid)	Synthetic polymer with mucoadhesive properties and biocompatibility	- Used in mucoadhesive microneedles for transmucosal drug delivery.
Poly (acrylic acid)	Synthetic polymer with pH-responsive properties and biocompatibility	- Used in pH-responsive hydrogel-forming microneedles for drug delivery.

**Table 2 materials-16-04805-t002:** Fabrication methods for hydrogel-forming microneedles as an example of polymeric microneedle (MN) production.

Method	Description	Advantages	Disadvantages
Micro-molding method	Fabrication using a micro-molding process to create MN structures made of hydrogel materials.	Precise control over MN dimensions Easy integration of drug encapsulation High reproducibility	Limited scalability Time-consuming process Difficulty in complex MN designs
Casting	Casting hydrogel material into MN molds	Simple and cost-effective fabrication Suitable for various hydrogel materials Scalable production	Limited control over MN geometry Potential for air bubble formation Difficulty in achieving uniformity
Electrospinning	Electrospinning hydrogel solution into MN structures	High flexibility in MN design Fine control over MN size and shape Enhanced mechanical strength	Complex and expensive equipment setup Limited drug loading capacity Difficulties in drug release control
3D printed hydrogel-filled microneedle array	3D printing of hydrogel-filled MN arrays	Precise control over MN geometry Ability to incorporate multiple drugs Customizable drug release profiles Potential for personalized medicine applications	Limited drug loading capacity Limited scalability Limited mechanical strength

## Data Availability

Not applicable.

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
