# Peer review of "Hydrogel-Forming Microneedles with Applications in Oral Diseases Management"

_materials, 2023, doi:10.3390/ma16134805_

Round 1
Reviewer 1 Report
Manuscript ID
Materials-2402383
Title of the Manuscript: Application of hydrogel-forming microneedle (HFM) in oral diagnosis and dental diseases management
In this review manuscript authors describe the application of hydrogel-forming microneedle (HFM) in oral diagnosis and dental disease management. This review article has value for the researchers in the related areas. However, the review article needs minor improvement before acceptance for publication.
My detailed comments are as follows:
1- Add a schematic diagram showing the different types of microneedles in addition to the mechanism in which the different microneedles are used for transdermal drug delivery.
2- Summary in Table the fabrication methods used for polymeric MNs showing the description of method, advantages, and disadvantages for all methods.
3- Summary in Table the materials used for hydrogel-forming microneedles.
4- Add a section showing the disadvantages of Hydrogel-Forming Microneedles.
5- Add a section showing the future direction of hydrogel-forming microneedles.
Thanks a lot
Minor editing of English language required
Author Response
Point 1: Add a schematic diagram showing the different types of microneedles in addition to the mechanism in which the different microneedles are used for transdermal drug delivery.
Response 1: The authors appreciated very much Revier 1’s suggestion. We now included a new schematic diagram (Figure 1) depicting the different types of microneedles in the revised manuscript.
Point 2: Summary in Table the fabrication methods used for polymeric MNs showing the description of method, advantages, and disadvantages for all methods.
Response 2: We have incorporated a summary table (Table 2) that outlines the hydrogel-forming microneedles fabrication methods as an example for polymeric MNs. This table includes a description of different methods, along with their respective advantages and disadvantages.
Point 3: Summary in Table the materials used for hydrogel-forming microneedles.
Response 3: Thank you for the comment, we now included Table 1 summarizing the materials commonly used for hydrogel-forming microneedles.
Point 4: Add a section showing the disadvantages of Hydrogel-Forming Microneedles.
Response 4:
We appreciate the reviewer's suggestion. To address this suggestion, we have revised the manuscript accordingly: We have divided the information about the “Advantages of hydrogel-forming microneedles in dentistry” into two distinct parts (now Sections 6 and 7) within the existing text. The first part focuses on the advantages (Section 6) related to MN, and the second part discusses the disadvantages (Section 7) associated with MN.
Point 5: Add a section showing the future direction of hydrogel-forming microneedles.
Response 5: We appreciate the reviewer's suggestion. Please note that we have already incorporated Section 8 in our manuscript, i.e. "Future perspectives": we discussed within the new section, various exciting applications and potential advancements for hydrogel-forming microneedles, including biosensing, accelerated dentoalveolar/periodontal wound healing, and anti-bacterial effects.
Point 6: Minor editing of English language required.
Response 6: Thanks for the comment. With the aid of the cloud-based software Grammarly, we have carefully reviewed our manuscript and made the necessary edits to improve the overall quality of the English language. We hope the current English standard is up to the Rviewer’s expectations.
Reviewer 2 Report
This is a very nice review paper on the application of the HMN applications. The authors started from the material, manufacture, and to the applications of HMN. While the flow of the reading is smooth, just a couple of comments for the authors.
(1) The abstract should be improved. It didn't capture a high-level summarize of the review paper. It is more like a small introduction.
(2) It would be nicer to add a content table to the review paper as it is now commonly seen. Also, this is because it takes patient to read up to section 5 to reach the core part of this review paper for HMN in dental applications. On the other hand, since each section is indeed quite short, the content table would give the readers better indices to reach each section of interested.
Very minor typo errors as far as I can see or care.
For example,
Line 729 on page 18, "is" sufficient quantity to "in" sufficient quantity.
Author Response
Point 1: The abstract should be improved. It didn't capture a high-level summarize of the review paper. It is more like a small introduction.
Response 1: Thank you for your critique regarding the abstract. Indeed we did not extensively revise the ‘proposed Abstract’ after the composition of the last manuscript version, apologies. We now revised it to provide a concise summary of the review paper, capturing the main findings and the essence of the review.
Point 2: It would be nicer to add a content table to the review paper as it is now commonly seen. Also, this is because it takes patient to read up to section 5 to reach the core part of this review paper for HMN in dental applications. On the other hand, since each section is indeed quite short, the content table would give the readers better indices to reach each section of interested.
Response 2: The authors appreciate the comment from Reviewer 2. A Table of Content is now inserted.
Point 3: Line 729 on page 18, "is" sufficient quantity to "in" sufficient quantity.
Response 3: We apologize for the typo. The error is revised.
Reviewer 3 Report
In this review article the authors have summarized the fabrication methods, biomaterials and disease management through hydrogel-forming microneedle (HFM). The literature search is thorough, the ideas are well presented, however there is a mismatch between title of the article and the depth of the relevant content with respect to oral disease and dentistry applications. Structural changes need to be introduced to align the content to the title by either condensing sections 1-3 and expanding section 4&5 OR keeping the current content but adjusting the title to the content. The main question addressed by the research reflected by the title " Application of hydrogel-forming microneedle (HFM) in oral diagnosis and dental diseases management " represents at most 35% of the article. 1) If the authors would like to keep the current title, they need to expand sections 4&5 and abbreviate the previous sections (1-3) only partially relevant to ORAL disease management. Hence a major restructuring is suggested. 2) If the authors wish to keep the current content, the title of the article needs to be REVISED. A suggestion would be "Hydrogel forming microneedles with applications in oral disease management". Since a wider scope is addressed then there was a gap in references that needed to be addressed: a) Cellulose deserves its own section in the carbohydrate-based microneedles.https://link.springer.com/article/10.1007/s10570-022-04859-1; https://www.ncbi.nlm.nih.gov/pmc/articles/PMC7564169/
b) The applications of microneedles in the introduction section and/or the disease management (Figure 3) should include microneedles and the emerging areas of microencapsulated cell delivery currently absent from the review: https://pubmed.ncbi.nlm.nih.gov/30065227/
Author Response
Point 1:
1) If the authors would like to keep the current title, they need to expand sections 4&5 and abbreviate the previous sections (1-3) only partially relevant to ORAL disease management. Hence a major restructuring is suggested.
2) If the authors wish to keep the current content, the title of the article needs to be REVISED. A suggestion would be "Hydrogel forming microneedles with applications in oral disease management". Since a wider scope is addressed then there was a gap in references that needed to be addressed:
Response 1: The authors appreciate very much the comments by Reviewer 3. The manuscript title is now revised as per the suggestion, keeping the current content. Many thanks.
Point 2:
- a) Cellulose deserves its own section in the carbohydrate-based microneedles.
https://link.springer.com/article/10.1007/s10570-022-04859-1; https://www.ncbi.nlm.nih.gov/pmc/articles/PMC7564169/
Response 2: Thanks for the comment. We apologize for the missed information. We now included a dedicated section on cellulose-based microneedles in the manuscript while citing the appropriate literature concerned Section 2.1.1.5 Cellulose [46-48].
Point 3:
- b) The applications of microneedles in the introduction section and/or the disease management (Figure 3) should include microneedles and the emerging areas of microencapsulated cell delivery currently absent from the review: https://pubmed.ncbi.nlm.nih.gov/30065227/
Response 3: The authors thank Reviewer 3 for the constructive comments. We have revised the introduction section (6th paragraph, [14]), current Figure 4 (previous Figure 3), Sections 4.2. (last sentence) and 4.2.6. to include the applications of microneedles in emerging areas, such as microencapsulated cell delivery and relevant references.
Reviewer 4 Report
1. Abstract: Provides overview & challenges of drug delivery in oral cavity. Reviewer appreciates concise intro.
2. Abstract: Highlights advantages of microneedles for controlled drug delivery. Emphasizes location specificity, low invasiveness, and retention.
3. Abstract: Introduces hydrogel-forming microneedles (HFM) and their mechanism of action for sustained drug delivery. Reviewer commends explanation.
4. Abstract: Mentions materials used in HFM fabrication, but lacks rationale and advantages. More info recommended.
5. Abstract: Highlights convenience of microneedle patches for painless drug delivery in mucosa. Important for clinical applications.
6. Abstract: Promises comprehensive review of HFMs in dentistry, but lacks specific findings. Adding examples or key outcomes would enhance impact.
7. Abstract: Mentions discussing limitations of HFMs, but lacks elaboration. Briefly mentioning challenges would provide balanced perspective.
8. Abstract: Well-written, but minor revisions suggested for clarity. E.g., rephrase sentence 3.
9. Abstract: Would benefit from concluding statement summarizing significance or potential impact.
10. Abstract: Meets requirements, but with minor improvements, it would engage readers and generate interest.
11. Q: Advantages of microneedles for transdermal drug delivery compared to oral administration or injections?
12. Q: Differences between natural and synthetic polymer hydrogels in properties and applications?
13. Q: More details on hydrogel microneedle fabrication process and key steps involved?
14. Q: Elaboration on hydrogel microneedle applications in dentistry and examples of use in treatments?
15. Q: Other types of natural polymers used in hydrogel microneedles and their advantages and applications?
16. GelMA hydrogel widely used in biomedical field. Used in wound healing, drug delivery, biosensing, tissue regeneration.
17. HAMA composite hydrogel with network structure, good properties. Used for managing diabetic ulcers, promoting angiogenesis, wound healing.
18. PVA water-soluble polar polymer with biocompatibility, wound healing properties. Used for large molecule delivery like siRNA.
19. Micro-molding, casting, electrospinning, 3D printing common methods for fabricating HFMs. Allow complex shapes, bioactive agent incorporation.
20. HFMs used for interstitial fluid extraction, diagnosing diseases. Show potential in cancer, diabetes, oral diseases.
21. More details on diagnostic tests performed with HFMs and advantages over traditional methods?
22. Evidence or references supporting HFM effectiveness in oral wound healing?
23. Elaboration on mechanism and advantages of antimicrobial-loaded hydrogel MNs compared to conventional treatments?
24. Recommendations for future research to evaluate efficacy and safety of HFMs in dental applications?
25. Ensure all abbreviations are defined upon first mention.
Author Response
Point 1: Abstract: Provides overview & challenges of drug delivery in oral cavity. Reviewer appreciates concise intro.
Response 1: Thank you for the advice. We revised the Abstract now to make sure the content is concise
Point 2: Abstract: Highlights advantages of microneedles for controlled drug delivery. Emphasizes location specificity, low invasiveness, and retention.
Response 2: Revised
Point 3: Abstract: Introduces hydrogel-forming microneedles (HFM) and their mechanism of action for sustained drug delivery. Reviewer commends explanation.
Response 3: The mechanism of actions explained
Point 4: Abstract: Mentions materials used in HFM fabrication, but lacks rationale and advantages. More info recommended.
Response 4: relevance of HFM materials updated.
Point 5: Abstract: Highlights convenience of microneedle patches for painless drug delivery in mucosa. Important for clinical applications.
Response 5: done
Point 6: Abstract: Promises comprehensive review of HFMs in dentistry, but lacks specific findings. Adding examples or key outcomes would enhance impact.
Response 6: key examples and outcomes were added.
Point 7: Abstract: Mentions discussing limitations of HFMs, but lacks elaboration. Briefly mentioning challenges would provide balanced perspective.
Response 7: limitations of HFMs elaborated.
Point 8: Abstract: Well-written, but minor revisions suggested for clarity. E.g., rephrase sentence 3.
Response 8: Done, thank you.
Point 9: Abstract: Would benefit from concluding statement summarizing significance or potential impact.
Response 9: Done, thank you.
Point 10: Abstract: Meets requirements, but with minor improvements, it would engage readers and generate interest.
Response 10: We appreciate your thorough evaluation of the abstract in our manuscript. We have carefully considered each of your comments and made extensive revisions to address the specific concerns raised.
Point 11: Q: Advantages of microneedles for transdermal drug delivery compared to oral administration or injections?
Response 11: Thank you for the query regarding the advantages of microneedles for transdermal drug delivery compared to oral administration or injections. We have carefully considered your comment and made revisions to our manuscript accordingly. In the revised version, we have included a dedicated paragraph to address this important aspect. The paragraph discusses the various advantages of microneedles (Section 6, third paragraph), such as enhanced bioavailability, non-invasiveness, reduced risk of needlestick injuries, improved patient compliance, versatility in customization, and potential for controlled drug release. These advantages position microneedles as a promising alternative for transdermal drug delivery, with the potential to revolutionize various clinical applications. We believe that these additions enhance the comprehensive nature of our review and highlight the unique benefits offered by microneedles in comparison to other delivery methods.
Point 12: Q: Differences between natural and synthetic polymer hydrogels in properties and applications?
Response 12: Thank you for your comment regarding the differences between natural and synthetic polymer hydrogels in properties and applications. In the revised manuscript, we put up a dedicated section to discuss the contrasting characteristics and applications of natural and synthetic polymer hydrogels (section 2.1, second paragraph).
Point 13: Q: More details on hydrogel microneedle fabrication process and key steps involved?
Response 13: Thank you for your valuable advice. Essential details are added concerning the fabrication process and key steps involved in hydrogel microneedle production are added to the paper Section 3, second paragraph, and Table 2).
Point 14: Q: Elaboration on hydrogel microneedle applications in dentistry and examples of use in treatments?
Response 14: Thank you for the suggestions. The manuscript was revised to elaborate more detailed HFM application in dentistry: Section 5.3, from line 10 to the end of the paragraph; and Section 5.7.
Point 15: Q: Other types of natural polymers used in hydrogel microneedles and their advantages and applications?
Response 15: Thank you very much for Reviwer 4’s comment/advice. We have revised the manuscript accordingly to provide a more comprehensive overview of natural polymers in hydrogel microneedles (Section 2.1, first paragraph). Specifically, we have included a section on cellulose-based hydrogel microneedles, along with their unique properties and potential applications in transdermal drug delivery (Section 2.1.1.5).
Point 16: GelMA hydrogel widely used in biomedical field. Used in wound healing, drug delivery, biosensing, tissue regeneration.
Response 16: Thank you for your valuable suggestion, and we have revised the manuscript accordingly to provide a comprehensive overview of GelMA's applications in the biomedical field, Section 2.1.1., first paragraph, line 10 from the top.
Point 17: HAMA composite hydrogel with network structure, good properties. Used for managing diabetic ulcers, promoting angiogenesis, wound healing.
Response 17: The comment is gratefully appreciated. Detailed information regarding HAMA’s good properties is added in Section 2.2.2. The applications are in Section 2.2.2., diabetic ulcer management, angiogenesis, and wound healing.
Point 18: PVA water-soluble polar polymer with biocompatibility, wound healing properties. Used for large molecule delivery like siRNA.
Response 18: The comment is gratefully appreciated. PVA’s positive characteristics are highlighted in Section 2.2.3.
Point 19: Micro-molding, casting, electrospinning, 3D printing common methods for fabricating HFMs. Allow complex shapes, bioactive agent incorporation.
Response 19: Yes, we have added a table (Table 1) summarising the fabrication methods to make the content more comprehensive. Thank you.
Point 20: HFMs used for interstitial fluid extraction, diagnosing diseases. Show potential in cancer, diabetes, oral diseases.
Response 20: The comment is gratefully appreciated. The applications of HFMs in medicine are highlighted in section 4, section 4.1. diagnosis and 4.2. diseases management.
Point 21: More details on diagnostic tests performed with HFMs and advantages over traditional methods?
Response 21: Grateful for the comment. We now incorporated the related details into the manuscript providing a comprehensive account regarding the diagnostic tests performed using HFMs and their advantages over traditional methods (Section 4.1.2, last sentence).
Point 22: Evidence or references supporting HFM effectiveness in oral wound healing?
Response 22: Apologies for the missed information. A recently published experimental study on oral wounds of diabetic rats showed that hydrogel adhesive could effectively protect against mucosal wounds and obviously shorten the inflammatory phase, thus promoting the wound-healing process [152]. The updated information is added to Section 8.2, second paragraph.
Point 23: Elaboration on mechanism and advantages of antimicrobial-loaded hydrogel MNs compared to conventional treatments?
Response 23: The potent antimicrobial properties of the antibiotic-loaded hydrogel-forming microneedles against both Escherichia coli and Staphylococcus aureus highlights the beneficial use of hydrogel-forming microneedles towards minimally invasive transdermal drug delivery of antibiotics [158].
Point 24: Recommendations for future research to evaluate efficacy and safety of HFMs in dental applications?
Response 24: Many thanks for the comment. A related section is added to address this issue (Section 8.4).
Point 25: Ensure all abbreviations are defined upon first mention.
Response 25: Sure, the issue is looked into and double-checked.
Reviewer 5 Report
The article report on “Application of hydrogel-forming microneedle (HFM) in oral diagnosis and dental diseases management” was carefully reviewed.
Wai Keung Leung Wai Keung Leung and co-authors demonstrate interesting work.
It needs revision before consideration for publication in the Materials.
The introduction still needs to be improved. The following references are relevant to the hydrogels and their biological applications, which should be accommodated in the introduction, result and discussion section to improve the quality of manuscript.
https://doi.org/10.1002/pat.5661
https://doi.org/10.1016/B978-0-323-90545-9.00005-7
https://doi.org/10.2174/2210303110666200206114632
The flow of the manuscript needs to be polished properly.
Some more innovative Figures to be considered.
Many important relevant references (3-5) from the Materials can be considered for the above-mentioned studies.
In the current state, there are some typographical errors. Therefore, the authors are advised to recheck the whole manuscript.
After addressing the above comments, the article may be considered for publication.
Revise
Author Response
Point 1: The introduction still needs to be improved.
Response 1: We have carefully revisited the introduction section and made significant improvements to enhance its clarity and effectiveness in setting the context. We added the emerging areas of microencapsulated cell delivery in the section and provided a more comprehensive overview of the background information. We have also incorporated relevant references to support the presented information. We believe that these revisions have strengthened the introduction and improved its overall quality. We thank you for bringing this to our attention and giving us the opportunity to enhance the manuscript.
Point 2: The following references are relevant to the hydrogels and their biological applications, which should be accommodated in the introduction, result and discussion section to improve the quality of manuscript.
https://doi.org/10.1002/pat.5661
https://doi.org/10.1016/B978-0-323-90545-9.00005-7
https://doi.org/10.2174/2210303110666200206114632
Response 2: We appreciate Reviewer 5’s suggestion and we have included additional references [50, 147, 149] related to hydrogels and their biological applications in the manuscript.
Point 3: The flow of the manuscript needs to be polished properly.
Response 3: Thank you for your valuable feedback. We have carefully revised the flow of the manuscript to ensure a smoother and more coherent structure. We have paid close attention to the organization of sections and paragraphs, ensuring logical transitions between ideas and concepts. Additionally, we have reviewed the language and sentence structure to enhance readability and clarity. We believe that these improvements have significantly polished the overall flow of the manuscript.
Point 4: Some more innovative Figures to be considered.
Response 4: Thank you for the suggestions. We have added one diagram and one table in the manuscript to enhance the readibility of this paper.
Point 5: Many important relevant references (3-5) from the Materials can be considered for the above-mentioned studies.
Response 5: Sure, we have added more relevant references. [77, 104, 107 and 145]
Point 6: In the current state, there are some typographical errors. Therefore, the authors are advised to recheck the whole manuscript.
Response 6: We apologize for any oversight in the manuscript and appreciate your feedback. We have thoroughly rechecked the entire manuscript to identify and correct any typographical errors that may have occurred. We also used the software Grammarly to minimize any difficult-to-recognize errors.
Point 7: After addressing the above comments, the article may be considered for publication.
Response 7: Thank you Reviewer 5’s critique and comments, appreciate that.